# Integrating Dietary Data into Microbiome Studies: A Step Forward for Nutri-Metaomics

**DOI:** 10.3390/nu13092978

**Published:** 2021-08-27

**Authors:** Francisca Yáñez, Zaida Soler, Manon Oliero, Zixuan Xie, Iñigo Oyarzun, Gerard Serrano-Gómez, Chaysavanh Manichanh

**Affiliations:** 1Gut Microbiome Group, Vall d’Hebron Institut de Recerca (VHIR), Vall d’Hebron Barcelona Hospital Campus, Passeig Vall d’Hebron 119-129, 08035 Barcelona, Spain; panchayanez@gmail.com (F.Y.); zaida.soler@vhir.org (Z.S.); manon.oliero@umontreal.ca (M.O.); xie.zixuan@vhir.org (Z.X.); inigooyarzunlafuente@gmail.com (I.O.); gerard.s.g@hotmail.com (G.S.-G.); 2Departament de Medicina, Universitat Autònoma de Barcelona, 08193 Barcelona, Spain; 3Gut Microbiome Group, Centro de Investigación Biomédica en Red de Enfermedades Hepáticas y Digestivas (CIBERehd), 28029 Madrid, Spain

**Keywords:** sFFQ development, relative validation, diet–microbiome relationship

## Abstract

Diet is recognised as the main driver of changes in gut microbiota. However, linking habitual dietary intake to microbiome composition and activity remains a challenge, leaving most microbiome studies with little or no dietary information. To fill this knowledge gap, we conducted two consecutive studies (*n* = 84: a first pilot study (*n* = 40) to build a web-based, semi-quantitative simplified FFQ (sFFQ) based on three 24-h dietary recalls (24HRs); a second study (*n* = 44) served to validate the newly developed sFFQ using three 24HRs as reference method and to relate gut microbiome profiling (16S rRNA gene) with the extracted dietary and lifestyle data. Relative validation analysis provided acceptable classification and agreement for 13 out of 24 (54%) food groups and 20 out of 29 nutrients (69%) based on intraclass correlation coefficient, cross-classification, Spearman’s correlation, Wilcoxon test, and Bland–Altman. Microbiome analysis showed that higher diversity was positively associated with age, vaginal birth, and intake of fruit. In contrast, microbial diversity was negatively associated with BMI, processed meats, ready-to-eat meals, sodium, and saturated fat. Our analysis also revealed a correlation between food groups or nutrients and microbial composition. Overall, we provide the first dietary assessment tool to be validated and correlated with microbiome data for population studies.

## 1. Introduction

For about 200,000 years, humans followed a hunter-gatherer lifestyle, using fire for cooking and eating wild game, fruits, vegetables, and nuts, with lipid, protein, and carbohydrate content each accounting for 33% of dietary intake [1,2]. Diet is recognized as the key driver of changes in the adult gut microbiota. Indeed, humans have co-evolved with their microbiota along with the development of agriculture [3,4]. The dietary profile of the modern human has changed rapidly over the last 100 years—much faster than normal evolutionary adaptation—such that it has undoubtedly had an impact on shaping our gut microbiota and, consequently, our health. Comparison of rural versus urban populations provides an interesting approach to understand the changes of the microbiome in the context of modern life. The faecal microbiota of individuals from two very different geographical locations with contrasting dietary habits, namely inhabitants of rural village in Burkina Faso and European children (EU), differ significantly with regard to the relative abundance of bacteria known to be involved in cellulose and xylan hydrolysis [5]. The urbanization of regions in China followed by geography, dietary habit, and ethnicity was shown to have an impact on the variation of the gut fungal microbiome, increasing *Sacc**haromyces cerevisiae* and depleting *Candida dubliniensis* [6].

Compelling evidence supports an association between changes in the microbiota community and human metabolic disorders, including obesity [7] and type 2 diabetes [8]. Moreover, many intervention studies have shown that diet alters gut microbiota. A shift from plant-based to animal-based diets has been linked to an increase in the abundance of bile-tolerant microorganisms and a decrease in Firmicutes, which metabolize dietary plant polysaccharides [9]. A relatively long-term diet intervention (12 months) showed that the Mediterranean diet improved cognitive function, frailty, and inflammation status in elderly individuals by modifying gut microbiota [10].

To the best of our knowledge, very few population studies have demonstrated an association between habitual diet, gut microbiota, and health status. Most studies related to microbiome profiling in the context of a specific disease or related to dietary intervention did not collect dietary data, while those assessing diet intake did not perform microbiome analysis [11,12]. We believe that the scarcity of studies relating diet, microbiome, and disease is not due to the lack of molecular or bioinformatics tools or even cost but to the unavailability of an appropriate dietary assessment tool that has been tested against microbiome data.

Along with 24-h dietary recalls (24HRs) and food records or diaries, food-frequency questionnaires (FFQ) are one of the most widely used dietary assessment tools. FFQs are usually designed in function of the purpose of the study and can contain from a few questions intended to seek the effect of a specific food or group of food [13,14] to a comprehensive list of between 5 and 350 items to capture habitual diet [15]. FFQs are considered one of the most suitable instruments for epidemiological studies compared to 24HRs or food records, as they are self-administered and therefore do not require the presence of a trained interviewer or considerable time dedication on the part of respondents. Moreover, they are cost-effective. However, FFQs, being usually very long, may lead to the misreporting of habitual dietary intake, and they have sometimes been reported to be unreliable. Additionally, current FFQs have been under-evaluated against microbiome data in epidemiological studies, as most microbiome studies contain few or no dietary data.

Here, we sought to (1) design a new semi-quantitative and simplified FFQ (sFFQ); (2) undertake a relative validation analysis; (3) perform a reproducibility analysis; and (4) correlate dietary intake with microbiome data.

## 2. Materials and Methods

### 2.1. Study Population

A total of 84 healthy volunteers (40 participants in the pilot study and 44 in the validation study) were recruited between May 2017 and August 2020 by disseminating an announcement. The study was conducted in accordance with Declaration of Helsinki, and the protocol was approved by the local Ethics Committee of the Vall d’Hebron University Hospital, Barcelona (Project identification code: PR(AG)156/2017). All participants signed a consent form.

Power calculation showed that a minimum of 40 subjects would be needed to give 85% power to detect correlation between sFFQ and mean three 24HRs of 0.45 as significant at the 5% level in order to take into account the small sample size. Exclusion criteria included age under 18 and over 65 years, antibiotic use during the three months prior to entering the study, use of proton pump inhibitor medication, and any disorders that may be associated with altered gut microbiota, such as diabetes, chronic digestive pathology, inflammatory bowel disease, and autoimmune disease.

### 2.2. 24-h Dietary Recall: The Pilot Study

We conducted a pilot study using three 24HRs on 40 healthy subjects to evaluate their dietary habits and select the food items to be added to new sFFQ. A dietitian and trained staff performed the three interviews, two of them during weekdays and one during the weekend. These interviews were used to collect data on food consumption on the previous day, from the first intake in the morning to the last meal and beverage consumed during the night. To avoid biases in the 24HR response, we conducted the interviews randomly in time, taking into account the participants’ availability. We assigned an alphanumeric code to each participant to maintain anonymity and registered the food in a 24HR collection form.

To objectively evaluate the serving size of each food and beverage, we used photographic albums: “Guide for dietary studies” from the Granada University [16] and the SU.VI.MAX. Portions Alimentaires [17].

To estimate the serving weight provided by the participant, we created a “Standardized household measures table” based on the “Food composition table” published by Moreiras et al. from the Complutense University of Madrid [18] and based on homemade food used for measuring purposes. For the latter, lab staff weighed their food, for which no information from any existing table or from the food industry was available, to recover an average weight for each one. To reduce possible bias introduced by the interviewers, we applied a standard operating procedure for data collection based on the five-step interview proposed by the USDA [19]. The mixed dishes were broken down into simple ingredients; for this, the participants were asked to describe the recipes and cooking procedures in detail.

To facilitate the search and reduce the encoding error for energy and nutrient quantification of food extracted from the 24HR, we created an inhouse food-composition database. For this purpose, we combined several food-composition databases, including the Spanish database (AESAN/BEDCA v 1.0, 2010), which includes a list of 950 foods and 31 nutrients (http://www.bedca.net; accessed on 10 March 2021); Moreiras’s table, which includes a list of 900 food items and nutrients [18]; and the USDA National Nutrient Database for Standard Reference [20], which has a list of 8618 foods and 150 nutrients. Our food-composition database currently contains 1104 foods and preparations grouped into 13 food categories and information for 29 nutrients plus energy per 100 g of food.

### 2.3. Design and Development of the sFFQ

To develop the sFFQ, we first applied a general concept following the recommendations from previous studies [15,21,22,23,24]. To select the food items to be included in the sFFQ, we then combined and cross-checked the data collected from the three 24HRs of our pilot study with the food-consumption data reported in the “National Food Survey on adults, the elderly, and pregnant women (ENALIA2)” [25] related to the foods most consumed by the Spanish population. In total, 310 foods were selected on the basis of their higher intake within the population and higher intra- and inter-individual variability of consumption. We added questions that could pinpoint relevant factors with a potential effect on microbiome composition changes, such as blood type [26], mode of delivery at birth [27], consumption of ready-to-eat meal [28], and whether or not the participant was following a specific diet [29] or was excluding a specific food or type of food. We also included other factors potentially associated with changes in the gut microbiota, such as age, BMI, smoking, use of sweeteners, and number of fruits and vegetables consumed [5,9,30,31,32,33,34,35,36,37].

Our resulting sFFQ contained 58 food items (Appendix A: List of 58 food items as specified in the sFFQ) in which the consumption frequency of the previous month was categorized into six possible responses for each item: “Never”, “1 or 3 times per month”, “1 or 2 times per week”, “3 or more times per week”, “once per day”, and “2 or more times per day”. We estimated the food serving size of each item of the sFFQ based on the results of various surveys and guidelines, including the ENALIA2 Survey [25], the guidelines of the Spanish Society of Community Nutrition [38], and the guidelines of the scientific committee “5 a day” [39]. We also used the serving size assigned by the food industry and the serving obtained from our own pilot study. To further improve estimation of the amounts of food consumed in the sFFQ, we created a support document based on food photographs and added three consumption alternatives for the standard serving size: “1/2 of the standard portion size”, “standard portion size” and “double the standard portion size” [40,41,42]. To estimate the energy and nutrient intake from the sFFQ, we used our food-composition database.

To recover the nutritional composition of the food items included in the sFFQ, we calculated the weighted mean of nutrients and energy based on the data obtained from the 24HR pilot study. First, the foods collected from this study were classified into 58 items, as presented in the sFFQ. We then recovered the proportions contributed by each food to each of the 58 items from the 24HR pilot study and used them as weighted factors to calculate the energy and nutrient intake for each item in the sFFQ.

### 2.4. sFFQ Administration

The sFFQ was used as a web-based survey using the SurveyMonkey Inc. (San Mateo, CA, USA) platform two times one month apart (sFFQ1 and sFFQ2). On the day of the first 24HR interview, we provided the participants with the web link or QR code to complete the sFFQ. Once we had obtained the responses, we verified the missing data. We contacted participants if there was a lack of response to any of the items in the sFFQ. A new version of the sFFQ using an independent online survey from our own server is currently being prepared.

### 2.5. Analysis of the sFFQ Responses

To compare the results of the sFFQ with the reference 24HR method, we transformed the monthly and weekly consumption data into daily consumption frequencies. To this end, we calculated the g/day as follows: a consumption response of 1 to 2 times per week was understood as an average consumption of 1.5 times per week, which, divided by the seven days of the week, gives an average daily consumption of 0.21. This consumption was then multiplied by the weight associated with the selected serving size (for example, for the legumes item with a serving size of 150 g and consumption frequency mentioned above, the final value of grams per day would be 0.21 × 150 g = 31.5 g/d). Using this g/day information, we then calculated the energy and nutritional value of each item in the sFFQ. Foods and beverages from the sFFQs and the 24HRs were then classified into 24 food groups, total energy, and 29 nutrients (Appendix A: Classification into food groups, energy and nutrients of the sFFQs and the mean 3–24HRs).

### 2.6. Identification of Unreliable sFFQs and 24HRs

We verified each value obtained from the quantification of the sFFQs and the 24HRs. When a possible outlier was detected, we examined each data entry involved in reaching this value. For instance, we excluded participants with calorie intake values in the sFFQ and means of 24HRs lower than 800 kcal/day or higher than 4200 kcal/day for men and less than 600 kcal/day or more than 3500 kcal/day for women [43,44].

### 2.7. Statistical Analysis to Evaluate the Validity and Reproducibility of the sFFQ

The median and 25–75 percentile of food, energy, and nutrient consumption were calculated from the mean of the three 24HRs and both sFFQs. Nutrients were adjusted by energy using the density method [45] to control the confounding effect of calories. The validity (sFFQ2 versus mean of the three 24HRs) and reproducibility (sFFQ1 versus sFFQ2) of the newly developed sFFQ were evaluated using a series of statistical tests.

To control for inter-and intra-individual variation, we calculated the intraclass correlation coefficient (ICC) [46,47]. We used cross-classification (CC) to categorize individuals into equal third or opposite third for food group and energy-adjusted nutrient intake extracted from both methods [48]. We used Spearman’s correlation coefficient to estimate the strength and direction of the association [22].

We applied the Wilcoxon signed-rank test to assess the differences in food, energy, and nutrient consumption and used the Bland–Altman analysis to check the degree of agreement between the two sFFQs and the three 24HRs. The differences between the two methods (FFQ2-24HR) were plotted against the mean intake of the measures ((FFQ2 + 24HR)/2) and the limits of the agreement, defined as the mean ± 1.96 SD of the mean between the two methods, were evaluated [49]. In addition, to illustrate the magnitude of the possible systematic difference, we calculated the 95% CI of the mean differences. To reflect the presence of proportional bias, Spearman’s correlation was calculated between the mean and the mean difference of the two methods [50]. Statistical analyses were performed in GraphPad Prism (v8) and the RStudio (Version 1.4.1106) package.

### 2.8. Microbiome and Statistical Analyses

Each of the 84 participants provided a faecal sample at baseline (M0), i.e., 24 to 48 h after the first 24HR, and one month after (M1), i.e., first stool after the third 24HR. Genomic DNA was extracted from 166 samples (two subjects did not provide a second sample), as previously described [51] and following the recommendations of the International Human Microbiome Standards (IHMS, http://www.human-microbiome.org/; accessed on 23 April 2021). The V4 hypervariable region of the 16S rRNA gene was PCR-amplified and sequenced using the MiSeq Illumina platform [52]. Sequence data were analysed using the QIIME 2^TM^, which is a bioinformatics platform that stands for Quantitative Insights into Microbial Ecology. The sequences were demultiplexed to attribute sequence reads to the appropriate samples and were then denoised and dereplicated into amplicon sequence variants (ASVs) using the dada2 tool, which also filtered out chimeras. Each sequence read was trimmed to a length of 298 bp. A total of 3.1 million sequences of the 16S rRNA gene were generated from the 166 samples, with a mean of 19,000 sequences per sample. A feature table was generated for all samples with a minimum of 9159 sequences per sample. One sample with a very low number of reads was removed for further analysis. The feature table of the 165 remaining samples was then used to perform taxonomic classification, alpha- and beta-diversity analyses, and differential abundance measurements in different experimental groups. Taxonomy was assigned to each ASV using a database that combined the Greengenes (version 13.8) and PATRIC (version 2016) databases. To study the association between the microbiome data and clinical or dietary variables, we then used linear mixed models as implemented in the Microbiome Multivariable Association with Linear Models (MaAsLin2) package [53]. MaAsLin2 was set up with the following parameters: normalization = “TMM”, transform = “LOG”, correction = “BH”, analysis_method = “LM”, max_significance = 0.25 (default significance threshold), min_abundance = 0.0001, min_prevalence = 0.1. Age, gender, and other characteristics of the participants as well as dietary data were added as fixed effects. All models were adjusted for gender, and as participant samples from two timepoints were included, the participant identification number was added as a random effect. Results with a false-discovery rate (FDR) lower than 0.25 were considered significant.

### 2.9. Deposition of Sequences Data

Sequence data have been deposited in the NCBI database with the following access number: PRJNA745527.

## 3. Results

### 3.1. Study Design

We built a new semi-quantitative and simplified food-frequency questionnaire (sFFQ) to assess usual dietary intake. This sFFQ would be useful for epidemiological studies seeking to correlate dietary information with microbiome data. To address this knowledge gap, we designed a pilot study (*n* = 40) to build a sFFQ based on the dietary habits of our population extracted from three 24HRs administered over a period of one month. From this pilot study, we generated a sFFQ, which included several lifestyle-related questions and 58 food items. The food items were classified into 24 food groups and 29 nutrients (Appendix A: Classification into food groups, energy and nutrients of the sFFQs and the mean 3–24HRs). Participants completed the sFFQ online.

To collect information on dietary intake, participants were interviewed three times by a trained technical staff over a period of one month in a form of three dietary recalls (24HRs) (baseline, day 15, day 30) and were asked to complete the online sFFQ at baseline (sFFQ1) and on day 30 (sFFQ2), as described in Figure 1 and in the method section. Participants collected stool samples at baseline and one month after (first stool after completing each sFFQ) and kept them in their home freezer until they could bring them to the lab, where they were kept at −80 °C.

### 3.2. Participants’ Characteristics

The participants (*n* = 84) in this study reside mainly in Spain, and 81% hold Spanish nationality. The cohort was recruited among staff from the Vall d’Hebron hospital, as well as their relatives and close friends, via flyers and word of mouth. Females accounted for 55.9% of the participants. The average age was 34.2 years old (from 20 to 64 years old), with 52.4% in the range of 18–29 years old. Among other relevant characteristics, 80% of the cohort presented a normal BMI (18.5–24.9 kg/m^2^), 85.7% were born vaginally, and 72.6% were non-smokers. Among the different types of diet, about 80% of the participants followed a conventional diet, but 60% reported the consumption of ready-to-eat meals and 25% the use of artificial sweeteners. Comparison analysis of several parameters, such as age, gender, BMI, nationality, type of birth, and dietary habits, between the pilot and the validation study did not reveal significant differences. More detailed information on the characteristics of the cohort is provided in Table 1.

### 3.3. Validation of the sFFQ

Dietary data extracted from the 24HRs were converted into 58 food items (Appendix A: List of 58 food items as specified in the sFFQ) as they were listed in the sFFQ, and all data were also converted into a list of 24 food groups, total energy, and 29 nutrients (Appendix A: Classification into food groups, energy and nutrients of the sFFQs and the mean 3–24HRs, Figure 2). The participants spent an average of 22 min answering questions on the 58 items (SD = 16.2 min, max = 86 min, min = 4 min).

To validate the newly developed sFFQ, we conducted a second study (*n* = 44)—a validation study—in which we compared the food items, food groups, and nutrients obtained from the sFFQ with the mean of the three 24HRs. For this comparison, we used data obtained from sFFQ2, which recalled the dietary intake of the previous month and therefore may better correspond to the mean of the three 24HRs of the same month.

We applied several statistical tests to measure the strength and direction of the association between the two different measurements at individual level (ICC, CC, and Spearman’s rank correlation tests) and to quantify agreement between the two measures at group level (Wilcoxon test and Bland–Altman plots), as recommended by Lombard et al. [22].

The median ICC coefficient of food groups between the sFFQ2 and the mean of the three 24HRs was 0.35 (range: 0.05–0.83), and the median ICC of energy-adjusted nutrients was 0.55 (range 0.08–0.98) (Appendix A: Relative validation of the sFFQ (sFFQ2 vs. mean 3–24HRs)). The cross-classification values satisfactorily classified participants on the basis of their intake based on the two methods since more than 50% of the participants were classified in the same tertile for 14 food groups and 13 nutrients, while less than 10% were classified for the opposite tertile in 12 food groups and 15 nutrients. The median Spearman’s correlation coefficient for food groups between sFFQ2 and mean of three 24HRs was 0.46 (range 0.18–0.78) and was also 0.46 (range 0.10–0.71) for energy-adjusted nutrients (Appendix A: Relative validation of the sFFQ (sFFQ2 vs. mean 3–24HRs)). Wilcoxon tests showed that 15 out of 24 (62.5%) food groups and 19 out of 29 nutrients (63%) were not significantly different between the two diet assessment methods.

Based on Bland–Altman analysis, we observed that the sFFQ2 tended to report a lower intake of biscuits breakfast cereals, chocolate and derivatives, pastries and sweet breads, ready-to-eat meals, sauces-condiments, and sausages and a higher consumption of the food groups of appetizers and vegetables than the 24HR based on the method used in Giavarina [50] (Appendix A: Relative validation of the sFFQ (sFFQ2 vs. mean 3–24HRs)). Regarding energy and nutrients, the sFFQ2 underestimated energy intake, total fat, polyunsaturated fatty acids (PUFA), saturated fatty acids (SFA), cholesterol, sodium, and selenium and overestimated vitamin D, folate, niacin, and vitamin C compared to the 24HR. Altogether, we observed that 13 out 24 (54%) food groups and 20 out of 29 nutrients (69%) were classified as good or acceptable according to the criteria reported in Lombard et al. [22].

### 3.4. Reproducibility of the sFFQ

We then evaluated the reproducibility of the sFFQ by comparing the dietary data extracted from the sFFQs administered on two occasions one month apart in the validation study. The median ICC coefficient of food groups was 0.63 (range: 0.21–0.90), and the ICC of energy-adjusted nutrients was 0.73 (range 0.58–0.96) (Appendix A: Reproducibility analysis of the sFFQ (sFFQ1 vs. sFFQ2)). The median Spearman’s correlation for food groups and energy-adjusted nutrients was 0.72 (range 0.32–0.90) and 0.70 (range 0.54–0.86) (Appendix A: Reproducibility analysis of the sFFQ (sFFQ1 vs. sFFQ2)), respectively. The Wilcoxon test revealed that no food groups or nutrients were significantly different between the two sFFQs.

### 3.5. Participants’ Dietary Profile

The food groups, energy, and nutrients extracted from the sFFQs and 24HRs were quantified (Figure 2 and Appendix A: Classification into food groups, energy and nutrients of the sFFQs and the mean 3–24HRs). The most consumed food groups in our population were vegetables (21%), non-alcoholic beverages (17%), fruits (14%), milk and dairy products except fermented milk (10%), alcoholic beverages (9%), meat and eggs (6%), fish and shellfish (4%), yoghurt and fermented milk (3%), and legumes (3%).

### 3.6. Correlation between Participants’ Characteristics and Microbial Diversity and Taxa

We collected two faecal samples from participants in the pilot (*n* = 40) and validation (*n* = 44) studies at baseline and one month later (first stool after each sFFQ). Microbiome composition was analysed based on the amplification and sequencing of the V4 region of 16S rRNA gene. We evaluated the association between microbial alpha-diversity (richness and evenness) or taxonomic profile (relative abundance of microbial genera) and several characteristics of the participants, including age, BMI, gender, smoking habit, blood type, and type of birth. To this end, we used linear mixed models implemented in the MaAsLin2 tool and took into account the longitudinal setting of the study, as the participant identification number (Subject ID) was added as a random effect.

Vaginal birth was found to be associated with higher microbial diversity (FDR < 0.02 for Chao1 (richness) and Shannon (evenness) indices) and resulted in enrichment in several bacterial genera, including an unclassified genus from the Ruminococcaceae family (FDR = 6.37 × 10^−7^), from the Clostridiales order (FDR = 5.80 × 10^−5^) and from RF39 (FDR = 0.0006) as compared with C-section births (Figure 3). Age was found positively correlated with diversity (FDR = 0.06 for Shannon index) and negatively correlated with *Bilophila* (FDR = 0.005). The pre-obese and obese group (BMI above 25) was associated with a lower microbial diversity (FDR = 0.018 for Chao1 and FDR = 0.12 for Shannon) and depleted in members of the Clostridiales order, such as *Facecalibacterium* (FDR = 0.10). BMI was classified following the World Health Organization’s recommendation as follows: underweight (BMI below 18.5), normal weight (BMI = 18.5–24.9), pre-obesity (BMI = 25.0–29.9), and obesity (BMI above 30.0). Smoking habit (smoker, non-smoker, or ex-smoker) was not associated with diversity or any microbial taxon. Gender was not found associated with diversity but was associated with a depletion of *Bilophila* (FDR = 0.04) in male participants. Use of sweeteners was negatively associated with *Desulfovibrio* (FDR = 0.07).

### 3.7. Correlation between Dietary Intake and Microbial Diversity and Taxa

To correlate microbiome data with dietary intake, we first used the 58 food items from the sFFQs (*n* = 44, 85 faecal samples; one sample did not provide sufficient high-quality sequences, and two subjects provided only one sample at baseline). The analysis was performed using MaAsLin2. The results are shown in Figure 4. Item 14, which consisted of fresh fruit, was positively associated with richness and evenness (FDR = 0.009 for Chao1 and Shannon) and with the relative abundance of a member of the Ruminococcaceae family (FDR = 0.1). Item 35, which consists of processed meats, was negatively associated with richness (Chao1, FDR = 0.034), evenness (Shannon, FDR = 0.03), and an unclassified genus from the Clostridiales order (FDR = 0.1). Item 58, which comprised process foods, was negatively associated with richness and evenness (FDR = 0.009 for Chao1 and Shannon) and with an unclassified genus from the Clostridiales order (FDR = 0.19).

We then correlated microbiome data with the 24 food groups extracted from the sFFQ. Fruits and fruit products, which encompassed food items 14, 15, and 16, were also found to be positively correlated with microbial diversity (FDR = 0.005 for Chao1 and Shannon indices). “Sausages and other processed meats” and “ready-to-eat meals”, which corresponded to food items 35 and 58, respectively, were found to be negatively correlated with microbial diversity and taxa, as mentioned above. No association was observed between microbiome data and the other items or food groups.

Finally, we did not uncover any association between the 29 nutrients and total energy extracted from the sFFQs and microbial diversity except for sodium (FDR = 0.005) (Figure 5) and saturated fatty acid (SFA) (FDR = 0.04) (Figure 5), whose levels were negatively correlated with both richness and evenness. Sodium was positively correlated with *Holdemania* and negatively correlated with *Ruminococcus* and *Methanobrevicter*, a member of the Ruminococcaceae family (FDR < 0.05). Other nutrients, such as alcohol, total fat, and total fibre, were also associated with several microbial genera (Figure 5). Alcohol was positively correlated with two genera from the Coriobacteriaceae family, one of them being *Collinsella* (FDR = 0.0004), and negatively correlated with a member of the Peptostreptococcaceae family (FDR = 0.03). Total fat and SFA were negatively correlated with *Ruminococcus;* total fat was also positively correlated with *Clostridium*. Monounsaturated fatty acids (MUFA) were negatively correlated with *Methanobrevibacter* (FDR = 0.03). Total fibre was positively correlated with a member of the Clostridiaceae family (FDR = 0.009).

## 4. Discussion

This study describes the development and validation of a semi-quantitative and simplified FFQ and the integration of demographic and dietary data into microbiome data. The newly developed online sFFQ, which contains 58 food items converted into 24 food groups and 29 nutrients, reports the dietary intake of an adult population in the last month. The reduced number of items was chosen to lessen the burden on respondents and to maximize their full attention. The time frame of one month is an attempt to match usual dietary consumption with changes in the microbiome community [54]. Furthermore, respondents completed the sFFQ in an average of 22 min, which is much less time than that needed for the most commonly used FFQs (from 30 to 60 min) [14]. We consider that a short FFQ will attract more volunteers who will be more willing to repeat the experiment several times in a year to cover, for instance, every season.

The Bland–Altman analysis reflected good levels of agreement between the sFFQ and DR, and the graphs showed that most of the data fell within the limits of agreement. The sFFQ reported 20 out of 29 (69%) nutrients and 13 out of 24 (54%) food groups with good or acceptable outcome compared to the 24HR reference method, as evaluated by at least three distinct statistical methods (Appendix A: Relative validation of the sFFQ (sFFQ2 vs. mean 3–24HRs). The sFFQ showed only 24% underestimation and 11% overestimation of the food, energy, and nutrient group. The results obtained for several food items (biscuits, breakfast, cereals, chocolate and derivatives, pastries and sweet breads, ready-to-eat meal, sauces-condiments, and sausages) for energy and for several nutrients, such as total fat, PUFA, SFA, cholesterol, sodium, and selenium, should be interpreted with caution since they were underreported, whereas vitamin D, folate, niacin, and vitamin C were overreported in the sFFQ compared to the three 24HRs. The under- or over-estimation trends of certain foods or nutrients could be explained by a series of characteristics of the participants and by social approval of certain foods [55,56]. Indeed, the consumption of foods considered “beneficial for health”, such as fruits and vegetables, are usually reported more frequently while that of “bad” foods, such as foods high in fat or sugar, are usually less frequently reported [57,58,59]. In addition, since the main objective of this dietary assessment was not to measure energy intake, the underestimation of energy should not impact the overall design of our sFFQ. At the group level, the correlation values for food groups (Spearman = 0.177–0.78 and ICC = 0.049–0.83) were within the ranges observed in previous validation studies in adults [60,61]. The energy-adjusted correlation values were similar to those reported in several validation studies [62,63]. The greatest discrepancies in cross-classification were observed especially for foods eaten sporadically (fish and shellfish, legumes, pastries, ready-to-eat meals, sausages), possibly due to the low probability of encountering these foods in the three 24HRs.

The very high correlation of our extracted data, with two published epidemiological studies (ANIBES and ENIDE) investigating the usual diet of the Spanish population, suggests that the newly built sFFQ could be applied at the population level (data not shown). Moreover, the very high repeatability of the questionnaire indicates that only one sFFQ would be needed to cover food intake over one month. However, to adapt the questionnaire to another population, validation using a reference such as a 24HR on a subpopulation would be needed to make additional and necessary changes to the sFFQ.

Our study, using three 24HRs on 84 healthy individuals, captured certain effects of BMI as well as lifestyle on the diversity and composition of the gut microbial community. The association between microbiome and demographic data was achieved with participants from both the pilot and validation study (*n* = 84, 165 faecal samples). The impact of type of birth on the gut microbial ecosystem has been widely studied during early life [64,65]. A persistent effect of mode of delivery on the microbiome composition, the host immune system [66], and the biosynthesis of natural antibiotics [67] has been reported during the first years of life. However, to the best of our knowledge, no relevant study has reported this effect in adult subjects. Our findings regarding relationship between age and diversity, depletion of Clostridiales, and overweight-obesity corroborate previous findings [68,69].

Contradictory findings have been published about the effect of sweeteners, such as saccharin, on glucose tolerance and dysbiosis in healthy individuals [70,71]. We observed that sweeteners, which consisted mainly of aspartame and saccharin in our study, decreased *Desulfovibrio*, which is a sulphate-reducing bacterium.

On the one hand, through the sFFQ, we were able to associate high microbial diversity, which is considered a health-promoting factor [72], with the intake of fruits and low diversity with processed meat, ready-to-eat meals, total fat, saturated fatty acids, and sodium intake. On the other hand, the sFFQ allowed us to correlate food items or nutrients with specific groups of microorganisms. Some members of the Clostridiales order were positively associated with fruits and total fibre, whereas others, including *Ruminococcus,* were negatively associated with total fat, saturated fatty acids, and sodium intake. The association with the latter should be interpreted with caution given that the comparison between sFFQ2 and the mean of the three 24HRs based on Bland–Altman showed that sodium was underreported by participants.

A low intake of dietary fibre has been related to loss of diversity and loss, in particular of members of Clostridiales [73] and the class Clostridia [74]. The non-association observed between vegetables and microbial diversity could be explained by the cooking method used. Indeed, fruits and vegetables may contain similar nutrients that could be lost during cooking procedures involving boiling, steaming, or stir-frying, methods commonly use to prepare vegetables. Cooking, which transforms fibre and starch, increasing their absorption in the small intestine and thus reducing their fraction in the colon, has been shown to reshape the structure and function of gut microbiota [75].

Our study showed that the newly developed sFFQ has the potential to capture the usual diet of adult healthy individuals as reflected by its validation with a reference method and a comparison with other epidemiological studies. Nevertheless, we need to stress several limitations. First, this study, which was observational in nature, requires, for instance, human interventional studies to further validate the associations found between diet and the microbial community. This study relies only on 16S rRNA analysis, which reveals only microbiome composition, and it could be complemented by functional analysis through DNA and RNA shotgun sequencing or metabolomics analysis. Moreover, the results obtained on several food items, energy, and several nutrients should be interpreted with caution since they have been shown to be misreported in the sFFQ compared to the three 24HRs. However, the 24-HR used here as a reference method, is not the gold standard. This reporting method also relies on memory and may be biased due to underestimation or overestimation. This limitation could be addressed by using metabolomic biomarkers, although only a few comprehensively validated biomarkers of food intake are available.

## 5. Conclusions

To the best of our knowledge, this newly developed sFFQ is the first to be validated and tested against microbiome data. This new sFFQ could be adapted and used in future population studies to assess diet in a population from another region of the world and/or to study the effect of diet and metabolic disorders. We expect this new tool to open up new avenues in both nutritional and microbiome fields leading to nutri-metaomics.

## Figures and Tables

**Figure 1 nutrients-13-02978-f001:**
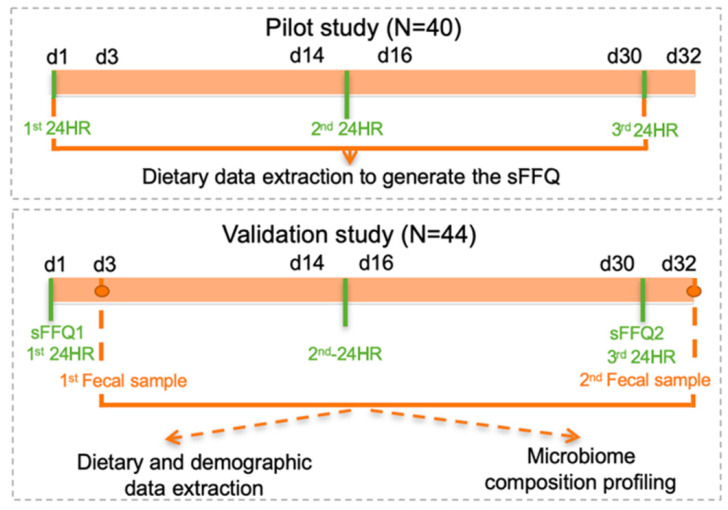
Study design. We first designed a pilot study to extract and quantify the dietary habits of our population. These data were then used to build a simplified food-frequency questionnaire (sFFQ). In the pilot study, participants were interviewed three times by trained staff members over a one-month period in a form of three dietary recalls (three 24HRs) (baseline, day 15, day 30), and in the validation study, they also underwent three 24HRs and were asked to complete two web-based sFFQs, one at baseline and the other a month later (first stool after completing each sFFQ). Participants provided a frozen stool sample on day 3 and day 32 for microbiome analysis.

**Figure 2 nutrients-13-02978-f002:**
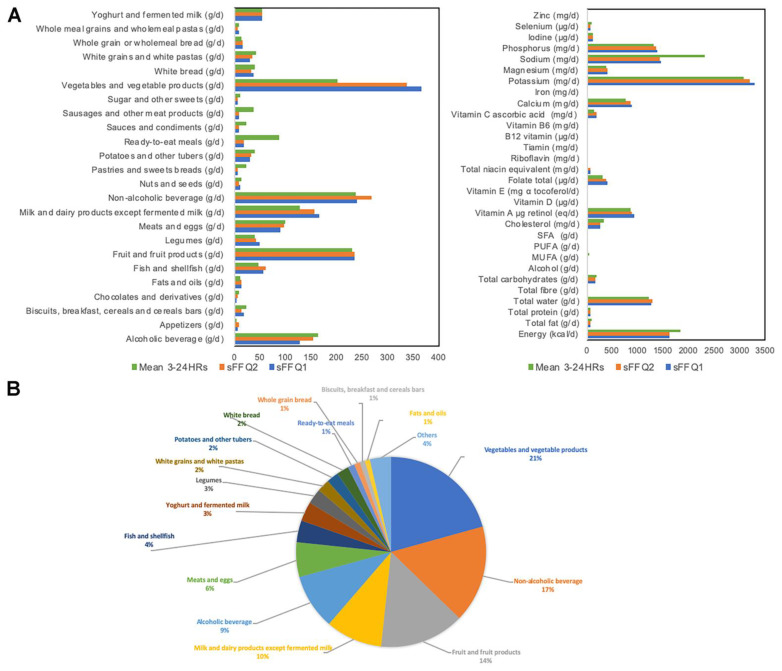
Participants’ dietary profile. (**A**) Food groups, energy, and nutrients extracted from the sFFQs and 24HRs. (**B**) Proportions of food groups as extracted from the sFFQ2 (validation study). Food groups with proportions lower than 1% were grouped into “Others”.

**Figure 3 nutrients-13-02978-f003:**
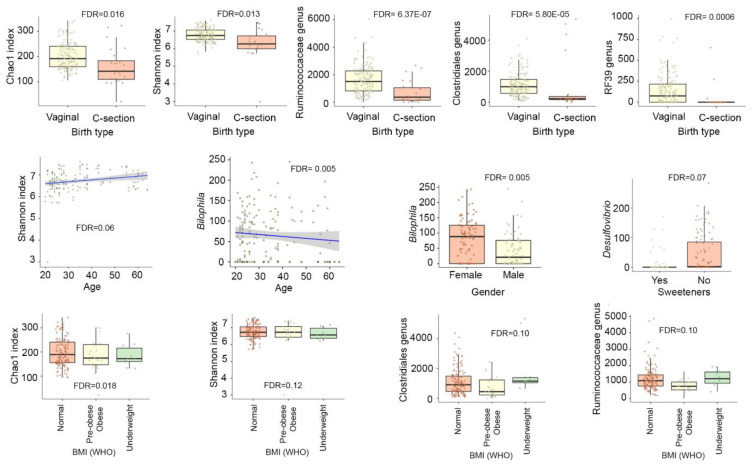
Association between participants’ characteristics and microbial diversity and taxa. Linear mixed models implemented in MaAsLin2 were used to analyse the microbiome and data on the characteristics of participants in the pilot and validation studies (*n* = 84, 165 faecal samples). BMI, body mass index; WHO, World Health Organization.

**Figure 4 nutrients-13-02978-f004:**
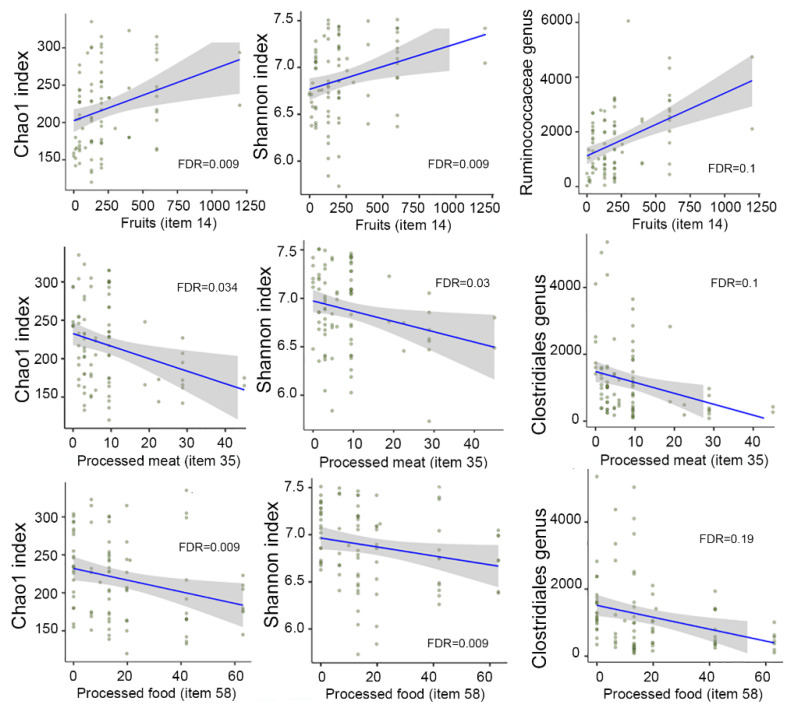
Correlation between food groups and microbial diversity and taxa. Linear mixed models implemented in MaAsLin2 were used to analyse the microbiome and dietary data extracted from the validation study (*n* = 44, 85 faecal samples).

**Figure 5 nutrients-13-02978-f005:**
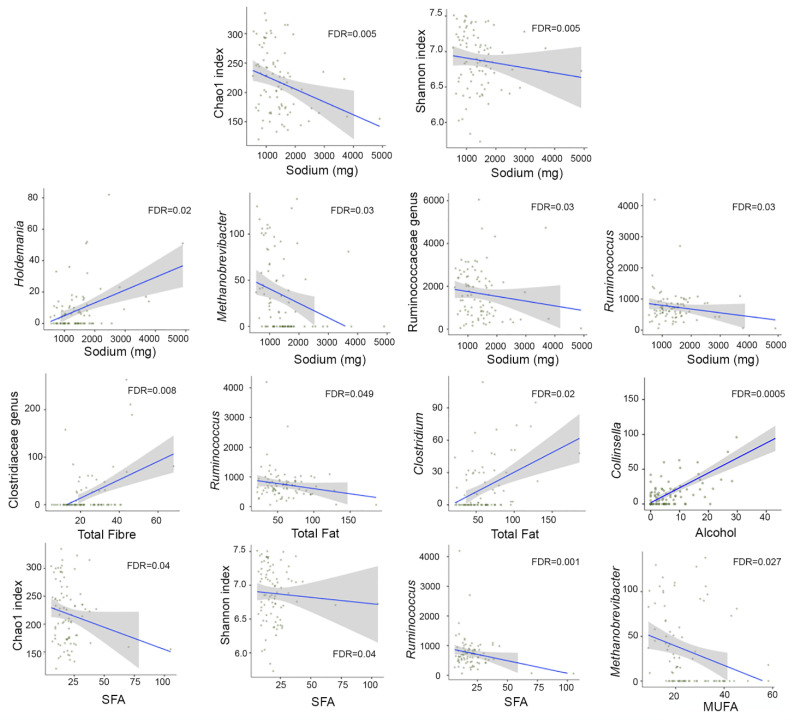
Correlation between nutrients and microbial diversity and taxa. Linear mixed models implemented in MaAsLin2 were used to analyse the microbiome diversity and taxonomic profile and nutrients data extracted from the validation study (*n* = 44, 85 faecal samples). SFA, saturated fatty acids; MUFA, monounsaturated fatty acids. All nutrients except for sodium (in mg) are displayed in grams.

**Table 1 nutrients-13-02978-t001:** Participants’ characteristics.

	Total	Pilot Study	Validation Study
*n*	84	40	44
Age (years)	34.2 ± 12.7	32.6 ± 11.1	35.7 ± 14.0
18–29 years, *n* (%)	44 (52.4)	21 (52.5)	23 (52.3)
30–39 years, *n* (%)	19 (22.6)	13 (32.5)	6 (13.6)
40–49 years, *n* (%)	7 (8.3)	2 (5.0)	5 (11.4)
50–59 years, *n* (%)	9 (10.7)	3 (7.5)	6 (13.6)
>60 years, *n* (%)	5 (4.2)	1 (2.5)	4 (9.1)
Female gender, *n* (%)	47 (55.9)	26 (65.0)	21 (47.7)
BMI (kg/m^2^)	22.5 ± 3.0	22.0 ± 2.6	23.1 ± 3.3
Weight status, *n* (%)			
Underweight (<18.5 kg/m^2^)	5 (4.2)	3 (7.5)	2 (4.5)
Normal (18.5–24.9 kg/m^2^)	67 (79.8)	35 (87.5)	32 (72.2)
Overweight (25–29.9 kg/m^2^)	10 (11.9)	2 (5.0)	8 (18.1)
Obese (>30 kg/m^2^)	2 (2.4)	0	2 (4.5)
Nationality, *n* (%)			
Spain	68 (81.0)	29 (72.5)	39 (88.6)
European—non-Spanish	8 (9.5)	6 (15)	2 (4.5)
Others	8 (9.5)	5 (12.5)	3 (6.8)
Birth type, *n* (%)			
Vaginal birth	72 (85.7)	35 (87.5)	37 (84.1)
C-section	12 (14.3)	5 (12.5)	7 (16.0)
Blood type, *n* (%)			
A	26 (30.9)	14 (16.7)	12 (14.3)
B	4 (4.7)	2 (2.4)	2 (2.4)
AB	1 (1.2)	0	1 (1.2)
O	33 (39.3)	15 (17.9)	18 (21.4)
Unknown	20 (23.8)	9 (10.7)	11 (13.1)
Smoking status, *n* (%)			
Non-smoker	61 (72.6)	32 (80)	29 (65.9)
Smoker	9 (10.7)	2 (5)	7 (15.9)
Former smoker	8 (9.5)	0	8 (18.2)
Unknown	6 (7.1)	6 (15)	0
Diet type, *n* (%)			
Conventional	67 (79.8)	31 (77.5)	37 (84.1)
Vegetarian diet	6 (7.1)	4 (10.0)	2 (4.5)
Vegan diet	2 (2.4)	1 (2.5)	1 (2.3)
Organic diet	2 (2.4)	2 (5.0)	0
Others diet	7 (8.3)	2 (5.0)	4 (9.1)
Intake of ready-to-eat meals, *n* (%)			
Yes	51 (60.7)	23 (57.5)	28 (63.6)
No	33 (39.3)	17 (42.5)	16 (36.4)
Intake of sweeteners, *n* (%)			
Yes	-	¯	11 (25.0)
No	-	¯	33 (75.0)
Intake of supplements or drugs, *n* (%)			
Dietary supplements	20 (23.8)	8 (20.0)	12 (27.3)
Probiotics	1 (1.2)	1 (2.5)	0
Oral contraceptive	6 (7.1)	4 (10.0)	2 (4.5)
ACE inhibidors	3 (3.6)	2 (5.0)	1 (2.3)
Fibrate	1 (1.2)	1 (2.5)	0
Statin	1 (1.2)	1 (2.5)	0
Levothyroxine	2 (2.4)	1 (2.5)	1 (2.3)
Other drugs	8 (9.5)	4 (10.0)	4 (9.1)

## Data Availability

Sequence data have been deposited in the NCBI database with the following access number: PRJNA745527.

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
