# Peer review of "Integrating Dietary Data into Microbiome Studies: A Step Forward for Nutri-Metaomics"

_nutrients, 2021, doi:10.3390/nu13092978_

Round 1

Reviewer 1 Report

The manuscript by Yañez et al. analyzes the correlation between food groups or nutrients and microbial composition.  The authors conducted two consecutive studies to build a web-based semi-quantitative simplified FFQ (sFFQ) based on three 24-hour dietary recalls and a second study to validate the newly developed sFFQ using 3DRs as reference method and to relate gut microbiome profiling with the extracted dietary and lifestyle data. The study generally supports the authors' conclusions. These findings may benefit from some additional clarification, as detailed below:

- Please indicate the clinical characteristic of the study population.

- What drugs are taken by the subjects? Antibiotics? Probiotics?

- The quality of the figures should be improved.

- The impact of the study should be better clarified and detailed.

- Waist circumference should be added and evaluated as a parameter in figure 3.

- The manuscript should be edited to correct contextual and layout errors.

Author Response

We wish to thank the referee for the time spent reviewing our paper. In this regard, we feel that all the points they raised have allowed us to improve the overall quality and readability of the manuscript. Our answers are written in red, below the comments made by the referee. The modifications made to the manuscript have been added with track changes.

We hope that you now find the new version suitable for publication.

REFEREE 1

The manuscript by Yañez et al. analyzes the correlation between food groups or nutrients and microbial composition.  The authors conducted two consecutive studies to build a web-based semi-quantitative simplified FFQ (sFFQ) based on three 24-hour dietary recalls and a second study to validate the newly developed sFFQ using 3DRs as reference method and to relate gut microbiome profiling with the extracted dietary and lifestyle data. The study generally supports the authors' conclusions. These findings may benefit from some additional clarification, as detailed below:

- Please indicate the clinical characteristic of the study population.

We appreciate the reviewer’s suggestion, which made us realize that we need to clarify that this study is currently designed to focus on a healthy population, but it could be adapted in the future to participants with diseases. However, the sFFQ would need to be accompanied by a questionnaire to collect specific clinical data adapted to the disease of interest. As recommended below, we have now clarified the impact of the study.

- What drugs are taken by the subjects? Antibiotics? Probiotics?

For the current project, the use of antibiotics was an exclusion criterion to enter the study. Indeed, antibiotics could have a strong effect on the gut microbiota community and we were aware that, with the cohort size planned for this study, we would not be able to properly evaluate the effect of distinct antibiotics taken by the participants. However, for a future larger project, this effect could be examined. We have now included the use of drugs and probiotics in Table 1. The number of subjects taking each drug or probiotic was too small to evaluate their effect on the gut microbiota.

- The quality of the figures should be improved.

We are not exactly sure what the reviewer means by the “quality of the figures”, whether this refers to conception, format, size, or resolution. We have made every effort to improve several of these points and we hope that the reviewer is now satisfied with the modifications.

- The impact of the study should be better clarified and detailed.

We appreciate and agree with the suggestion and have added the following text:

“…This new sFFQ could be adapted and used in future population studies to assess diet in a population from another region of the world and/or to study the effect of diet and metabolic disorders…”

- Waist circumference should be added and evaluated as a parameter in figure 3.

Waist circumference was not measured for the participants. Neither was weight or height. These latter data were self-reported. We might have misled the reader using the term “anthropometric” when referring to the information collected. We have now replaced this term with BMI.

“…Our study, using 3-24HRs on 84 healthy individuals, captured certain effects of BMI, as well as lifestyle, on the diversity and composition of the gut microbial community…”

- The manuscript should be edited to correct contextual and layout errors.

We thank the reviewer for drawing our attention to these issues. We believe that we have made the changes necessary, although part of the formatting was done by the editorial team.

Reviewer 2 Report

Nutrients (ISSN 2072-6643)

nutrients-1316694

Manuscript ID

«Integrating dietary data into microbiome studies: a step 3 forward for Nutri-metaomics»

The autors are linking  habitual dietary intake to microbiome compositions are linking  . They conducted  two consecutive studies (n=84: a first pilot study (n=40) to build a web-based semi-quantitative  simplified FFQ (sFFQ) based on three 24-hour dietary recalls; a second study (n=44) served to  validate the newly developed sFFQ using 3DRs as reference method and to relate gut microbiome  profiling (16S rRNA gene) with the extracted dietary and lifestyle data.

Microbiome analysis  showed that higher diversity was positively associated with age, vaginal birth, and intake of fruit. In contrast, microbial diversity was negatively associated with BMI, processed meats, ready-to-eat  meals, sodium and saturated fat. Their analysis also revealed correlations between food groups , nutrients and microbial composition

Overall comment: Overall this is a well performed study. Using FFQs is complicated, and by validating  a new short FFQ is, as the reviewers have done in this study, makes it much easier to recruit patients in planned studies.

 Comments: Title: The title tells me that the present study is a part of larger studies in future metaomics studies. This could be mentioned in «future studies»

Blood tests and endoscopy and biopsies data are not part of the present study

Comments:

  1. Inflammation. If the data in the present study will be used in clinical studies, inflammation will be an important aspect. To do so measures of caprotectin i faeces could be wise.
  2. Short chain fatty acids. To examine the role of microbiota and fermentation of carbohydrates into SCFA in faeces could also have been done
  3. Figure 2a and 2b, Fig 5. Fiber and vegetables are large parts of the diet and there seems to be significant correllations to thefood groups and microbioal taxa
  4. FODMAP. Did any of the particpants use low FODMAP diet and is it possiblle to do correlations to high and low FOFMAP in the present data?
  5. Cooking methods. This part is well discussed
  6. Vaginal birth is well discussed
  7. The article has to be shortened, it is to long and there are several paragraphs that are repititons
  8. The results of the study can be used in Spain and southern part of Europe, but not so easily in US and Northern part of Europe, the design, however, and the new «short FFQ» could be used by other groups.

Author Response

We wish to thank the referee for the time spent reviewing our paper. In this regard, we feel that all the points they raised have allowed us to improve the overall quality and readability of the manuscript. Our answers are written in red, below the comments made by the referee. The modifications made to the manuscript have been added with track-changes.

We hope that you now find the new version suitable for publication.

REFEREEE 2:

The autors are linking  habitual dietary intake to microbiome compositions are linking  . They conducted  two consecutive studies (n=84: a first pilot study (n=40) to build a web-based semi-quantitative  simplified FFQ (sFFQ) based on three 24-hour dietary recalls; a second study (n=44) served to  validate the newly developed sFFQ using 3DRs as reference method and to relate gut microbiome  profiling (16S rRNA gene) with the extracted dietary and lifestyle data.

Microbiome analysis  showed that higher diversity was positively associated with age, vaginal birth, and intake of fruit. In contrast, microbial diversity was negatively associated with BMI, processed meats, ready-to-eat  meals, sodium and saturated fat. Their analysis also revealed correlations between food groups , nutrients and microbial composition

Overall comment: Overall this is a well performed study. Using FFQs is complicated, and by validating  a new short FFQ is, as the reviewers have done in this study, makes it much easier to recruit patients in planned studies.

Comments: Title: The title tells me that the present study is a part of larger studies in future metaomics studies. This could be mentioned in «future studies»

Response: We have now included the following text in the conclusion section:

“…This new sFFQ could be adapted and used in future population studies to assess diet in a population from another region of the world and/or to study the effect of diet and metabolic disorders…”

Blood tests and endoscopy and biopsies data are not part of the present study

Response: As we answered the first reviewer: “We appreciate the reviewer’s suggestion, which made us realize that we need to clarify that this study is currently designed to focus on a healthy population, but it could be adapted in the future to participants with diseases. However, the sFFQ would need to be accompanied by a questionnaire to collect specific clinical data adapted to the disease of interest. As recommended above we have now clarified the impact of the study”

“…This new sFFQ could be adapted and used in future population studies to assess diet in a population from another region of the world and/or to study the effect of diet and metabolic disorders…”

Comments:

  1. If the data in the present study will be used in clinical studies, inflammation will be an important aspect. To do so measures of caprotectin i faeces could be wise.

Response: As our current study sought to understand the effect of diet on the microbiome in a healthy population, we did not include any question in the sFFQ related to disease. However, we agree with the reviewer that measures of calprotectin in faeces to evaluate the effect of inflammation would be wise in future clinical studies. However, to do so, the sFFQ would need to be accompanied by a questionnaire to collect specific clinical data adapted to the disease of interest.

  1. Short chain fatty acids. To examine the role of microbiota and fermentation of carbohydrates into SCFA in faeces could also have been done

Response: We agree with the reviewer and thank him/her for this suggestion. We have now planned to develop other experiments using a metabolomic approach to examine the role of the microbiota on the production of SCFA in faeces. However, since we would not be able to include the results of these experiments in the present manuscript in a reasonable time frame, they will be the focus of another manuscript.

  1. Figure 2a and 2b, Fig 5. Fiber and vegetables are large parts of the diet and there seems to be significant correllations to thefood groups and microbioal taxa

Response: Yes, we found an interesting correlation between fiber and bacterial taxa. This association could be used in future studies to further evaluate the association between these taxa and the fiber ingested.

  1. Did any of the particpants use low FODMAP diet and is it possiblle to do correlations to high and low FOFMAP in the present data?

Response: Unfortunately, we do not consider our sFFQ suitable to evaluate the effect of FODMAP on the microbiome, since it would require dietary records or 24-dietary recalls to collect specific information, such as the content of polyols in processed foods, the maturity of fruits, and the cooking method used.

  1. Cooking methods. This part is well discussed

Response: We appreciate the comment.

  1. Vaginal birth is well discussed

Response: We appreciate the comment.

  1. The article has to be shortened, it is to long and there are several paragraphs that are repititons.

Response: We appreciate the comment and have now removed several repeated parts of the manuscript.

  1. The results of the study can be used in Spain and southern part of Europe, but not so easily in US and Northern part of Europe, the design, however, and the new «short FFQ» could be used by other groups.

Response: We agree with this statement.

Reviewer 3 Report

  • Very intriguing title and important field of study: idea of relating FFQ with microbiome is novel
  • FFQ is a very attractive tool for dietary assessment but much more difficult to develop one: usually very much tailored to specific needs (and/or specific target population)
  • 2 important contents in this paper: one for FFQ development and another with validation + microbiome
  • Current contents are not suitable for 1 paper: usually development & validation for separate papers or development & validation of FFQ in one paper. Validation using microbiome should be done separately after full validation of FFQ: In current form, adding microbiome distracts focus
  • 'DR' usually denotes dietary records. For 24-hr recalls, '24HR' is used internationally that using 'DR' for 24-hr recalls causes confusion
  • For 24HR, representative & reliable recipe database is necessary and critical but that is not mentioned/handled anywhere in the paper
  • It is hard to expect 24HR is done properly and that draws question on validation of FFQ especially because the results of 24HR in this study may not be reliable
  • Authors even questioned the usefulness of 24HR, which they have used as a reference (control), at the end of paper after all the analysis for validation 
  • Accordingly, discussion on microbiome data in relation with FFQ results may mean nothing
  • For FFQ also, more descriptive explanation for the whole process of development and much larger number of subjects are necessary to be qualified for publication

Minor comments

  • Line 110: reference 17 is wrongly cited
  • Line 174-175: What do you mean by ‘……. normalized the information obtained from the sFFQs…’? How can you normalize information?
  • Line 296: ‘4 minutes spent in answering FFQ’ leads to question of sincerity in response
  • Line 344-346: comparison with other study results which used totally different methods, databases and subjects makes no sense. No reference on ENIDE among reference No. 56-62
  • Microbiome profile needs to be related with foods, not individual nutrients
  • Line 486-487: I am not sure if anyone can say so because there is no evidence in this paper
  • Line 509-510: many vegetables do not have enough starch to be transformed to begin with

Author Response

We wish to thank the referee for the time spent reviewing our paper. In this regard, we feel that all the points they raised have allowed us to improve the overall quality and readability of the manuscript. Our answers are written in red, below the comments made by the referee. The modifications made to the manuscript have been added with track changes.

We hope that you now find the new version suitable for publication.

REFEREE 3:

  • Very intriguing title and important field of study: idea of relating FFQ with microbiome is novel

Response: We appreciate the comment.

  • FFQ is a very attractive tool for dietary assessment but much more difficult to develop one: usually very much tailored to specific needs (and/or specific target population)

Response: We agree with this statement. It took us about four years to develop our sFFQ.

  • 2 important contents in this paper: one for FFQ development and another with validation + microbiome

Response: This is correct.

  • Current contents are not suitable for 1 paper: usually development & validation for separate papers or development & validation of FFQ in one paper. Validation using microbiome should be done separately after full validation of FFQ: In current form, adding microbiome distracts focus

Response: We are aware that most studies related to FFQ development do not provide validation of the tool developed. However, we believe that undertaking this validation after the development of our sFFQ allows for a more complete and comprehensive study and a first necessary step to show that we can relate dietary data with microbiome structure and composition, which was the ultimate goal of our study.

We are also aware that this work requires additional adaptation or validation using larger population cohorts from another region of the world where dietary habits may be significantly different from that of our population, or by recruiting participants with a specific disease.

  • 'DR' usually denotes dietary records. For 24-hr recalls, '24HR' is used internationally that using 'DR' for 24-hr recalls causes confusion

Response: Following the suggestion, we have replaced DR by 24HR throughout the manuscript.

  • For 24HR, representative & reliable recipe database is necessary and critical but that is not mentioned/handled anywhere in the paper

Response: To the best of our knowledge, there is no standard recipe database for Spanish dishes. However, in order to reduce biases as much as possible, we took several measures, such as:

- asking restaurants where the participants had eaten the dishes for the recipes.

- calculating the nutritional composition based on the Spanish nutrition labeling exercise book regarding the recipes provided by the participants.

(https://www.aesan.gob.es/AECOSAN/docs/documentos/seguridad_alimentaria/gestion_riesgos/labelling_nutrition_tolerances_1212_en.pdf).

-taking into consideration the edible and non-gross weight of the food and water loss percentage according to the type of cooking methods used based on tables provided by the National Food and Nutrition Center of Peru, available to us.

(https://www.academia.edu/41736564/II_TABLA_DE_FACTORES_DE_CONVERSI%C3%93N_DE_PESO_DE_ALIMENTOS_COCIDOS_A_CRUDOS).

  • It is hard to expect 24HR is done properly and that draws question on validation of FFQ especially because the results of 24HR in this study may not be reliable
  • Authors even questioned the usefulness of 24HR, which they have used as a reference (control), at the end of paper after all the analysis for validation 
  • Accordingly, discussion on microbiome data in relation with FFQ results may mean nothing

Response to the three comments above: 24HR is commonly used as a reference method to validate the development of new sFFQs, but we are aware that 24HR may have flaws and is often not considered the gold standard. However, we believe that we have taken all the precautions necessary to optimize the interviews, as indicated in the manuscript in the following lines:

  • 100-101 “A dietitian and trained staff performed the three interviews”
  • 104-105 “To avoid biases in the 24h-DR response, we conducted the interviews randomly in time, taking into account the participants' availability”
  • 108-110: “To objectively evaluate the serving size of each food and beverage, we used photographic albums: “Guide for dietary studies” from Granada University [16] and the SU.VI.MAX. photographic Atlas”
  • 116-118 . “To reduce possible bias introduced by the interviewers, we applied a standard operating procedure for data collection based on the five-step interview proposed by the USDA”

Questioning the usefulness of 24HR was not our first intention, we simply wanted to pinpoint to the fact that the validation of our sFFQ based on an 24HR is a relative validation, since the 24HR is not a gold standard. Further validation of our tool would be required by designing, for instance, a population/epidemiological study with a much larger cohort. We have now included the impact of the study in more detail as follow:

“…This new sFFQ could be adapted and used in future population studies to assess diet in a population from another region of the world and/or to study the effect of diet and metabolic disorders…”

  • For FFQ also, more descriptive explanation for the whole process of development and much larger number of subjects are necessary to be qualified for publication

Response: We have added the following text to section: 2.3. Design and development of the sFFQ

“… In total, 310 foods were selected on the basis of their higher intake within the population and higher intra- and inter-individual variability of consumption. We added questions that could pinpoint relevant factors with a potential effect on microbiome composition changes such as blood type [26], mode of delivery at birth [27], consumption of ready-to-eat meals [28], and whether or not the participant was following a specific diet [29] or was excluding a specific food or type of food”.

Minor comments

  • Line 110: reference 17 is wrongly cited

Response: This has now been corrected.

  • Line 174-175: What do you mean by ‘……. normalized the information obtained from the sFFQs…’? How can you normalize information?

Response: We have corrected this as follow:

“…Foods and beverages from the sFFQs and the 24HRs were then classified into 24 food groups, total energy, and 29 nutrients…”

  • Line 296: ‘4 minutes spent in answering FFQ’ leads to question of sincerity in response

Response: Since we cannot verify the sincerity of response, we could not remove this participant. However, we noticed that the same participant was consistent in answering the other sFFQ, as it took him 6 min and we found that his responses were sound, which suggests that he could be just a very fast reader.

  • Line 344-346: comparison with other study results which used totally different methods, databases and subjects makes no sense. No reference on ENIDE among reference No. 56-62

Response: We have decided to remove the comparison from this study.

  • Microbiome profile needs to be related with foods, not individual nutrients

Response: As we believe that there are no strong arguments in the literature against relating microbiome with nutrients and since we are performing an exploratory study, we related microbiome profiles with both nutrients and food groups, as shown in Figure 4.

  • Line 486-487: I am not sure if anyone can say so because there is no evidence in this paper

Response: We understand the skepticism of the reviewer regarding our speculation. We have therefore decided to remove this statement from the manuscript.

  • Line 509-510: many vegetables do not have enough starch to be transformed to begin with

Response: The non-association observed between vegetables and microbial diversity could be explained by the cooking method used. Indeed, fruits and vegetables may contain similar nutrients (bioactive compounds derived from plants, such as polyphenols), which could be lost or altered during the cooking procedures used to prepare vegetables.

Round 2

Reviewer 1 Report

no comment